# Preliminary Assessment of Intramuscular Depot of Lipid-Based Decoquinate Formulation for Long-Term Chemoprophylaxis of Malaria

**DOI:** 10.3390/pharmaceutics14122813

**Published:** 2022-12-15

**Authors:** Yinzhou Fan, Li Qin, Zhenping Huang, Shuanghong Liang, Xiaoyi Huang, Sumei Zeng, Yucheng Liu, Hongxing Wang

**Affiliations:** 1Division of Nanomedicine, Bluelight Pharmatech, Co., LTD, Science City, Guangzhou 510530, China; 2CAS Lamvac Biotech Co., Animal Service, Guangzhou 510670, China; 3Guangzhou Institute of Biomedicine and Health, Chinese Academy of Sciences, Guangzhou 510530, China; 4Laboratory of Nanomedicine and Biomaterials, School of Biomedical Science and Engineering, South China University of Technology, Guangzhou International Campus, Guangzhou 430030, China; 5Department of Biochemistry and Molecular Biology, Tongji Medical College, Huazhong University of Science and Technology, Wuhan 430074, China

**Keywords:** sustained release, decoquinate, *Plasmodium* infection, cholesterol

## Abstract

Sustained-release formulations of decoquinate were evaluated for the long-term prophylaxis of malaria. In the initial experiment, mice were protected from liver-stage *Plasmodium* infection by intramuscular administration of a lipids-based formulation at a dose of decoquinate 200 mg/kg. The mice that were inoculated with *Plasmodium berghei* sporozoites 34 days after the administration of a one-time drug dose were continuously monitored for 60 days and shown to be free of *Plasmodium* parasites. The optimized formulation for the sustained release of decoquinate was prepared by hot melt extrusion, constructed by lipids including cholesterol and mono or diglycerides, and had a drug load of 20 to 40% and particle size of 30 to 50 μm. Decoquinate of the lipids-based formulation was slowly released in vitro at a constant rate for the duration of two months, and was examined and continuously exposed at a therapeutic level in the blood for as long as 4 to 6 months. Further evaluation showed that the lipids-based formulation at doses of decoquinate 100 to 150 mg/kg could protect mice from *Plasmodium* infection for a period of 120 days. It is the first time that cholesterol has been used for a controlled drug delivery system of decoquinate. The results may provide useful information, not only for preparing a formulation of long-acting decoquinate but also in general for developing a controlled drug release system. The one-time administration of pharmaceutical agents in such a slow-release system may serve patients with no concerns about compliance.

## 1. Introduction

The World Health Organization (WHO) reported that there were an estimated 241 million cases of malaria in 2020, and estimated malaria deaths of 627,000, about 80% of which were children under the age of 5. The incidence of malaria is increasing in some regions. In 2020, about half of the world’s population was at risk of malaria. Most cases and deaths occur in sub-Saharan Africa. However, the other WHO regions also report significant numbers of cases and deaths [1].

*Plasmodium* sporozoites enter the hepatocytes of susceptible human beings when an infected mosquito bites. The sporozoites injected from the salivary glands of mosquitos are moved on into the blood circulation through which they travel to the liver and invade the hepatic parenchymal cells where sporozoites develop into liver-stage schizonts. *Plasmodium* (*P*) *falciparum* and *P. vivax* are the two most common parasites causing human malaria. In human primary hepatocytes, the development of *P. falciparum* takes 7 days, whereas *P. vivax* completes development after 9 days [2]. Some parasites such as *P. vivax* and *P. oval* can have their sporozoites differentiated into both liver-stage schizonts and hypnozoites. The schizonts then burst to release uninucleate merozoites. Hepatocyte-derived merozoites invade erythrocytes and enter a few-step cycle to develop into schizonts-containing asexual cells. Certain host erythrocytes develop into gametocytes associated with sexual cells. The hypnozoites are the dormant parasites in the liver that can cause a relapse of malaria months or even years later. The liver stage provides an ideal target for antimalarial intervention to block malarial progression before the manifestation of clinical symptoms caused by the rupture of schizonts from erythrocytes [3,4].

Antimalarial medications can target the *Plasmodium* liver stage and/or blood stage. Artemisinin, acting on blood-stage *Plasmodium*, is the first-line drug for treating severe malaria caused by *P. falciparum*. It must be combined with a different class of antimalarials to avoid drug resistance. Artemisinin has not been recommended for malarial prophylaxis. Chloroquine is prescribed for the treatment of *P. vivax* malaria and is no longer recommended for treating *P. falciparum* malaria. Chloroquine does not have activity against liver stage *Plasmodium*. It is prescribed for the intra-erythrocytic stages to prevent malaria only in a few areas with chloroquine-sensitive malaria [5]. It is taken by travelers one to two weeks before departure and continued for four weeks after return. Mefloquine and doxycycline are also blood schizontocidal agents. Mefloquine is taken at least one week before departure and continues for 4 weeks after return. It has more side effects than chloroquine. Doxycycline is taken daily for travelers visiting chloroquine-resistant or mefloquine-resistant areas. These medications require frequent intake and good compliance of travelers with physicians’ prescriptions.

There are several antimalarials that target the *Plasmodium* liver stage, including atovaquone/proguanil (Malarone), primaquine, and tafenoquine. Malarone is two combined drugs with synergistic actions that can be taken as a causal and suppressive prophylactic agent, but it is not effective in suppressing hypnozoites. Therefore, it cannot be used as a radical cure for malarial relapse caused by *P. vivax* or *P. oval*. It is, due to its cost, generally limited to travelers from industrialized countries. Primaquine and its derivative tafenoquine, a recently approved new drug, are the only drugs effective against hypnozoites. However, both primaquine and tafenoquine can cause fatal hemolysis in people with G6PD deficiency and should never be prescribed as prophylaxis to anyone with G6PD deficiency or unknown G6PD status [5]. Tafenoquine has two dosage forms, one (Arakoda) used for the chemoprophylaxis of malaria in individuals traveling in endemic areas, and the other (Krintafel) as a radical cure of *P. vivax* in patients ≥ 16 years. Krintafel should only be used in combination with chloroquine, not other antimalarials, against acute *P. vivax* infection.

As a measure of prophylaxis, vulnerable populations take antimalarial drugs at regular intervals to prevent acute malaria attacks. Currently, all drugs used for prophylaxis require frequent administration. There have been efforts to make controlled-release formulations of antimalarials to treat experimental cerebral malaria or blood-stage *Plasmodium* so that the activity of active pharmaceutical ingredients (API) can last for a longer period [6,7,8]. However, there have been no reports of such slow-release formulations of liver-stage drugs for long-term causal chemoprophylaxis of malaria.

In addition to chemoprophylaxis, another important aspect of controlling malaria involves monoclonal antibodies applicable to fight against malaria. Recently, a long-acting antimalarial monoclonal antibody CIS43LS was assessed in a two-part, phase 1 clinical trial [9]. Participants underwent controlled human malaria infection in which they were exposed to mosquitoes carrying *P. falciparum* sporozoites 4 to 36 weeks after the administration of CIS43LS. The monoclonal antibody provided causal prophylaxis, which targets *Plasmodium* infection at the intrahepatic stage, and the results were encouraging. The only vaccine currently available is RTS, S/AS01, which prevents infection, maturation, and multiplication within the liver but has limited efficacy [10]. It has been used in vulnerable populations rather than travelers via the Implementation Program [5]. Although numerous studies are ongoing to identify novel candidate antigens, there are presently no vaccines available for travelers.

One alternative that has been in progress is the use of long-acting injectable formulations of API [11]. Long-acting injectable formulations are depot delivery systems intended for prolonged/sustained drug release over a prolonged period ranging from a few days to months. Better patient compliance may be achieved through the parenteral route of administration over the oral route for the treatment of many chronic and life-threatening diseases. Parenteral depots may also reduce the relapse rate of disease and the maintenance phase of therapy, hence improving efficacy and treatment adherence. Examples of long-acting injectables reaching commercial viability so far are the depot products mostly for antipsychotic, substance abuse, and hormonal therapy drugs [12].

A great deal of research aiming at application has been conducted by using biodegradable polymers. Poly (lactic-co-glycolic acid) (PLGA) is the most widely used polymeric biomaterial in controlled drug delivery systems including small chemical and large biopharmaceutical molecules [13,14]. However, the use of PLGA for the purpose of prolonged release may not meet the requirements of every agent in various cases, such as controlling encapsulated drug release rates and/or formulation instability. Drug delivery systems with high efficiency and sustained-release characteristics involve drug load and release profiles [11].

Decoquinate (DQ) is a potent antimalarial agent targeting multistage *Plasmodium* parasites [15,16]. As with atovaquone, its mechanism of action is to inhibit the cytochrome bc1 complex in the mitochondria. However, there was little cross-resistance between the two agents because they block different sites [17]. As an anti-coccidiostat used for years, DQ has an excellent safety profile [18]. DQ can effectively inhibit chloroquine-sensitive *P. falciparum* as well as chloroquine- or multidrug-resistant *P. falciparum* in infected human erythrocytes and protect mice from severe malaria [19,20]. To solve the problem of water insolubility and to improve bioavailability, DQ has been made into nanoparticles without compromising its antimalarial potency as an oral dosage form targeting liver stage *Plasmodium* infection. The nanoparticle formulations of DQ are very potent at inhibiting *Plasmodium berghei* in vitro at the liver stage (IC50 < 0.5 nM) and highly efficacious (1–3 mg/kg) in protecting mice from liver stage malaria [20,21]. The in vivo efficacy of DQ against liver stage *Plasmodium* might also be associated with its pharmacokinetic feature in that there is high enrichment of DQ in the liver whether the nanoparticles are orally or intravenously administered [22].

The lipophilic property may make DQ an appropriate candidate as a sustained-release formulation for the chemoprophylaxis of malaria. Recently, Li et al. created formulations of DQ in oil-based carriers to provide extended efficacy through continuous drug release over a prolonged period from a single injection of the depot to effectively protect mice from *Plasmodium* infection [23]. Interestingly, the microparticle formulation provided a 2.2-fold longer drug exposure and a 3–4 times longer prophylactic effect than the nanoparticle formulation. Although there have been experiments performed in animals to prove that DQ has excellent antimalarial activity, it has not become a candidate for clinical trials to evaluate its safety and efficacy in preventing or treating malaria. Long-acting injectable atovaquone nanomedicines for malaria prophylaxis have also been reported [24], but nanoparticles may not be as sustainable as microparticles for drug release [23].

Hot melt extrusion (HME) technology has been applied for pharmaceutical formulations to improve the dissolution rate and bioavailability of poorly water-soluble drugs and to make a sustained drug release of API over an extended period [25]. HME has been shown to be a suitable, highly efficient method to prepare the oral dosage form of DQ targeting liver stage *Plasmodium* infection [20]. In this study, various preparations of DQ as a prophylaxis of malaria were made by different methods including HME, and evaluated in vitro and in mice. A lipids-based DQ depot was stored in the muscle and provided a long-lasting release of antimalarial activity against *Plasmodium* infection.

## 2. Materials and Methods

Decoquinate (Pharmaceutical grade, purity ≥ 98.0%) was purchased from Genebest Pharmaceutical Co. Ltd. (Zhejiang, China). Decoquinate standard (Cat. No. 1165408, LOT F0G036) was bought from USP Rockville, MD, USA. Mono or diglycerides of medium-chain fatty acids (Capmul^®^ MCM) were from IMCD (Abitec, Old Dixie Hwy, Miami, Florida, USA). Cholesterol was from Beijing Dingguo Changsheng Biotechnology Co., Ltd. (Beijing, China). Pharma 11 HME machine was manufactured by ThermoFisher Scientific (Thermo Electron (Karlsruhe) GmbH, Germany). PLGA 75/25 (75% polylactic acid and 25% polyglycolic acid; Mw, 72,000) and PLGA 50/50 (50% polylactic acid and 50% polyglycolic acid; Mw, 18,000) were bought from Jinan Daigang Biomaterial Co., Ltd. (Shandong, China). N-methyl-2-pyrrolidone (NMP) was from Sigma-Aldrich ((Shanghai) Trading Co., Ltd.). All other chemicals and solvents were of analytical grade. For DQ quantitation, HPLC grade acetonitrile, ethanol, and methanol were obtained from Tianjin Kemiou Chemical Reagent Company (Tianjin, China).

C57 mice were purchased from Southern Medical University, China, and employed to assess formulation efficacy, drug concentration in the blood (pharmacokinetics), and formulation toxicity. Animal studies were conducted in strict accordance with the relevant animal testing regulations (Animal Management Regulations, PR China, issued on 14 November 1988, revised version on 1 March 2007). All experimental protocols involving animals were carefully examined and approved on 4 September 2017 by the Animal Care and Ethical Committee of CAS Lamvac Biotech Co., Guangzhou, China (ZKLH/BL 0904.2017-P10, 4 September 2017).

### 2.1. Preparation of Sustained-Release Formulations of Decoquinate

Sustained-release formulation of DQ (SRFD) was prepared by different methods but the method eventually selected was HME. The percentage of each component was accounted for by weight. Cholesterol (40–90%) was fully mixed with DQ (10–40%). Then mono and diglycerides of medium-chain fatty acids (Capmul MCM) (5–20%) were added to the above mixture. The thoroughly mixed ingredients were loaded into the HME machine. The temperature for the HME process ranged from 70 °C to 100 °C, and the screw speed was 50 rpm. The extrudate was either made as a solid stick of a given length for subcutaneous (SQ) injection or pulverized into a powder and then suspended in normal saline (pH5.5) for intramuscular injection.

The PLGA 75/25 or 50/50 (10–90%) was formulated into the SRFD either via HME or by being dissolved in NMP. For the HME process, the temperature ranged from 80 °C to 135 °C, and the screw speed ranged from 50 rpm to 100 rpm. For the non-HME process, PLGA 75/25 (50%) was first dissolved in NMP, and then DQ (50%) was added to the PLGA solution. The mixture was stirred for 2 h at 200 rpm at room temperature. The mixed composition was then solidified and evaluated for the in vitro release of DQ. PLGA 75/25 (50%) alone could not be formulated with DQ via the HME process. Therefore it was combined with cholesterol and processed via HME. As a result, it was incorporated into a formulation as a minor carrier of DQ (Table 1, F4, and 5). As shown in Table 1, formulations 1 to 8 were processed by the HME method, whereas the emulsion method was used to prepare F9 by adding cholesterol and DQ to 5 mL triglycerides (Captex 300), heating at 80 °C degrees, and stirring to form DQ emulsion. F10 was prepared by simply dissolving DQ in organic solvent NMP. Particle sizes of SRFDs were measured by Beckman LS 13320 particle analyzer. Some evaluated SRFD preparations are shown in Table 1.

### 2.2. X-ray Diffraction Analysis

SRFD was analyzed by X-ray diffractometer (XRD, Empyrean) at the Test Center of Sun Yat-sen University in Guangzhou, China. XRD is used for observing the crystal structures of chemical compounds. The shining of an X-ray on a crystal generates a diffraction pattern characteristic of the chemical structure. A powder form of the material can be analyzed. Crystalline substance generates their characteristic sharp peaks. To obtain information on the physical property of SRFD, XRD patterns of HME product, the physical mixture of the formulation prior to the HME process, and pure DQ were observed by the standard operation rules. The targeting material used was copper with CuK α radiation, and settings were regularly used conditions.

### 2.3. Drug Release Test

One gram of SRFD sample was placed in a 10 mL tube filled with 6 mL phosphate-buffered saline (PBS, pH 7.4), with shaking speed at 100 rpm under 37 °C. DQ released from tested SRFDs was measured by HPLC 1260 analysis (Agilent) and compared to the total amount of the drug from each sample used in the tests. Samples were taken periodically and continuously for 52 days. The cumulative release rates of DQ against time duration were calculated by dividing the amount of the released drug measured at each time point over the total amount of the drug placed. The detected drug each time (day) is the amount of all drugs released and accumulated from preceding times.

### 2.4. Drug Load Determination

Depot formulations were suspended in saline or PBS. The suspensions were mixed with 3 times the volume of ethanol in Eppendorf tubes and vortexed for 2 min. After standing for half an hour, the extraction was centrifuged for 5 min at 5500× *g* (Eppendorf Centrifuge 5810R, Hamburg, Germany). The ethanolic supernatants were taken for DQ quantification by a high-performance liquid chromatography (HPLC) system (Agilent 1260, Santa Clara, CA, USA) (AGILENT TECHNOLOGIES (CHINA) CO. LTD, Chaoyang, Beijing) and the drug load of each SRFD was calculated. Experiments were performed in triplicate for all formulations.

### 2.5. Decoquinate Quantitation 

DQ was quantitated by HPLC using a Diamonsil C18 column (250 × 4.6 mm, 5 µm; Beijing Dikma Technologies, Inc. China). All the in vitro and in vivo experiments including drug concentrations measured in the blood, in vivo efficacy studies, drug load determination, and drug release tests, were based on HPLC quantitation to calculate the amount of drug used or determined. For instance, in an efficacy experiment for a mouse weighing 25 g at a dose of DQ 200 mg/kg, an SRFD used was determined by HPLC to contain 5 mg DQ/12.5 mg total weight (40% drug load).

The wavelength of ultraviolet detection in HPLC was 260 nm. The isocratic mobile phase was absolute ethanol/H2O (80:20, 0.1% formic acid in both solutions). The column temperature was set to 35 °C, flow rate 1 mL/min, and injection volume 20 μL. All chemicals and solvents were of analytical grade. HPLC grade acetonitrile, ethanol, and methanol were obtained from Tianjin Kemiou Chemical Reagent Company (Tianjin, China).

### 2.6. Decoquinate Depot for Plasmodium Prophylaxis in Mice

C57 mice were employed to assess the chemoprophylaxis efficacy of SRFD in *Plasmodium* infection. Animal studies were conducted in strict accordance with the relevant animal testing regulations (National Regulations of Laboratory Animal Management, PR China, amended on 31 October 2017).

Seven-week-old C57 mice, one in each cage, were kept for at least 7 days after arrival before starting the experiments. Room temperature was 18–26 °C, the relative humidity was 34–68%, and the light and darkness alternately cycled for 12 h. Standard feeding food was provided before and during the study. Sporozoites (SPZ) of *P. berghei* ANKA expressing firefly luciferase were isolated from the salivary glands of laboratory-reared female Anopheles stephensi mosquitoes. The mosquitoes were maintained at 26 °C for 17 to 22 days after feeding on *Plasmodium* parasites from infected Kunming mice. Salivary glands separated from malaria-infected mosquitoes were manually broken to obtain SPZ. For assessment of the prophylactic efficacy of SRFD targeting hepatic *Plasmodium* infection, 50,000 SPZ per mouse were intravenously injected via the tail vein.

HME extrudates were dosed either in solid sticks given to mice by SC to each side of hind legs by using trocar needles or in liquid form administered to mice by IM after being pulverized into a powder and suspended in saline. The liquid form for administration by intramuscular injection was made by suspending the HME extrudates in normal saline with pH 5.5. The solid stick with a diameter of about 2 mm was shaped by HME output and the length of the stick was determined by weight based on the drug content equivalent to the amount of drug administered. The liquid suspension was given to both hind legs to cut the total volume in half to alleviate discomfort.

Liver stage infection of *P. berghei* was monitored by using the in vivo imaging system (IVIS). The images were obtained at 24 h, 48 h, and 72 h post SPZ inoculation. *P. berghei* infecting mice reside in the liver for intrahepatic development in the first 48 h and then spread from the liver to the whole body after 48 h. This time course is reflected in images characteristic of bioluminescent signals in different regions of the body. The signals indicative of the location of parasites expressing firefly luciferase were generated by giving the mice 150 mg/kg luciferin (Gold Biotechnology, St. Louis, MO) intraperitoneally. Meanwhile, fluorescent flux counts were also acquired to represent quantitative data. The counts measured were bioluminescent photons emitted from whole bodies or regions of intensity (ROI). The ROI settings of the Living Image^®^ 4.0 software reflect the luminescent intensity. A 3-D bioluminescent imaging tomography was performed with the software to obtain sequential images with filters ranging from 580 to 660 nm.

Blood-stage *Plasmodium* infection was monitored by microscopic examination of either a thin film or a thick film. Blood smears were stained with 3% Giemsa for 20 min. Parasite density was estimated by counting parasites against RBCs on the thin film:

Parasite density per μL = Number of parasites counted × RBC count per μL ÷ Number of RBCs counted. Infected red blood cells per 10,000 red blood cells were counted in thin films. In the cases where parasitemia was undetectable on the thin film, the thick film was prepared to count parasites against WBCs:

Parasite density per μL = Number of parasites counted × WBC count per μL ÷ Number of WBCs counted. Infected red blood cells per 100 WBCs were obtained. Survival rates were calculated from the date of *Plasmodium* SPZ inoculation to the endpoint of the efficacy of each SRFD, to determine the time lengths of SRFD protection in various dosages.

### 2.7. Measurement of Decoquinate Concentrations in the Blood

Propranolol 5% in purity, purchased from Sigma, St. Louis, MO. USA, was used as an internal standard. DQ standards were made by dissolving it in ethanol at 50 μg/mL as stock solution, which was then diluted to a series of 10 gradient concentrations from 0.5 ng/mL to 500 ng/mL. Each standard (20 μL) was mixed with blank blood 100 μL in a 1.5 mL tube by vortex for 3 min. Protein precipitation solution 400 μL (ethanol /acetonitrile (1:1) containing propranolol 1000 ng/mL) was added and then mixed again for 5 min. After standing at room temperature for 8 h, the tubes were vortexed for 5 min and centrifuged at 16,000× *g* for 60 min at 4 °C. The supernatant 100 μL was transferred to each well of the 96-well plate and DQ was measured by liquid chromatography–mass spectrometry (LC-MS) system (Applied Biosystems-SCIEX model for API 3000 mass spectrometer).

To measure DQ in the blood, the whole blood sample (100 μL) was mixed with 20 μL ethanol for 3 min. Precipitation solution (400 μL) was added and mixed for another 5 min. The rest of the procedure was the same as that for standard curve preparation. Data were analyzed by one-way analysis of variance (ANOVA) and expressed as mean ± SD.

The same species of mice (C57) as those used in efficacy experiments was dosed with F2 at DQ 200 mg/kg by IM. After the drug administration, blood samples were collected from mice by cardiac puncture on different days (day 1, day 5, day 15, day 30, day 60, day 90, day 120, day 150, and day 180) with a group of 5 mice for each time point. Due to the limited sample size, DQ was measured by LC-MS, only one point per animal sample. Original data points were plotted to represent the drug level in the blood on that day.

## 3. Results

### 3.1. Long-Lasting In Vitro Release of Decoquinate

DQ dissolutions of the SRFD are illustrated in Figure 1A. When PLGA, a biodegradable carrier, was used as a major carrier for the sustained delivery of DQ produced by HME, no DQ could be detected in the dissolution media (data not shown). When PLGA was used as a minor carrier and cholesterol as a major carrier (F4 and F5), the formulation released only 5.4% and 5.6% of the DQ, respectively, over 52 days, whereas F2, F3, F6, and F7, those that did not contain PLGA, released 25 to 35% of DQ over the same duration.

### 3.2. Validation of Sustained-Release Formulations of Decoquinate 

Chemoprophylactic efficacy on malarial infection was evaluated for a variety of formulations at a dose of DQ 200 mg/kg. Mice were inoculated with *Plasmodium* SPZ 34 days after SRFD placement. The parasitemia was examined 4 days after the inoculation. F1, a cholesterol-based SRFD, and both F2 and F3, a cholesterol–MCM-based SRFD, were suspended in saline and administered to mice by IM. As shown in Figure 1B, all three formulations were completely protective as mice in these groups had no parasitemia. Thick films were prepared to detect parasitemia because parasites in these mice were initially undetectable on thin films. Since neither thin films nor thick films showed any parasites, no parasitemia in these mice could be concluded. Mice in all three groups survived *berghei* malaria by day 60 (Figure 1C).

Compared to the liquid form of the SRFD (F1, F2, and F3), F3 and F4 in the solid stick (implant) given to mice by SQ had only partial effectiveness in preventing mice from *Plasmodium* infection. F9 composed of DQ and cholesterol in an oily medium prepared by emulsion formation was shown to have better efficacy than the solid SRFD (F3 and F4) in suppressing *Plasmodium* infection in mice but was not completely protective (Figure 1C).

### 3.3. In Vivo Efficacy of Lipids-Based Decoquinate Formulation

Cholesterol–MCM-based SRFD (F3) and SRF (F3 excipients or F8) were suspended in saline for IM injection and compared in the efficacy experiments. The SRFD (F10), generated by dissolving PLGA (L/G 75/25) with an equal amount of DQ in NMP, was also assessed. Primaquine (PQ) phosphate was administered at 30 mg/kg to mice by IM. F3 and F10 were dosed at DQ 200 mg/kg. Six weeks later, the mice were inoculated with 50,000 SPZ of *Plasmodium berghei*. The experiment design is shown in Figure 2A. The images acquired by in vivo image system (IVIS) 46 h after the SPZ inoculation showed the effectiveness of each IM placement in preventing parasite infections in mice. As shown in Figure 2B, F10 did not protect mice as effectively as F3. The photon counts obtained from the same data of IVIS, corresponding to the images, also confirmed the results (Figure 2C). The individual parasitemia in each different group were examined 6 days after SPZ inoculation (Figure 2D), and then the experiments were followed up by monitoring parasitemia for 60 days (Figure 2E). Survival rates on day 60 after the SPZ inoculation (Figure 2F) indicated that mice administered with F10 could not survive *berghei* malaria by day 30. Thus, when PLGA was formulated for sustained-release, whether acting as a minor carrier (F4) in the solid form (Figure 1B) or as a major carrier (F10) in the liquid form (Figure 2C), did not release sufficient DQ for the prophylaxis of *Plasmodium* infection in mice.

PLGA with L/G 50/50 was evaluated in preliminary experiments but was less effective in suppressing *Plasmodium* infection than PLGA L/G 75/25. Therefore, the subtype L/G 75/25 was used. Polylactic acid (PLA), polyglycolic acid (PGA), and polycaprolactone (PCL) were also assessed as excipients of DQ depot formulations, which did not turn out to be sufficient in suppressing *Plasmodium* infection (data not shown). There could be more validation work necessary for PLGA and these other polymers.

F3 was shown to be completely effective in preventing mice from *Plasmodium* infection (Figure 2C), results consistent with the data shown in Figure 1B,C. Interestingly, one animal placed with the F3 depot had a parasitemia peak on day 15, which then gradually disappeared on day 35 (Figure 2E), indicating that the infected mouse could have recovered by a continuous release of DQ from the SRFD or by assistance with its own immunity. All five mice with F3 survived the *Plasmodium* infection 60 days after the SPZ inoculation.

PQ served as a positive control. The API was not made into a formulation for sustained drug release and therefore could not be used in high doses. Such a low dose (30 mg/kg) was only partially effective against *Plasmodium* infection with little protection (Figure 2F). The formulation made of excipients (F8) was ineffective in preventing *Plasmodium* infection.

### 3.4. Physicochemical Properties of Lipids-Based Decoquinate Formulation

The X-ray diffractogram showed two peaks characteristic of pure DQ at the angles 3.77°2θ and 7.53°2θ (Figure 3A). Similar patterns, but with peaks much shorter, appeared in the F2 physical mixture (Figure 3B) compared with those of pure DQ. The F2 physical mixture had the same formulation components of F2 prior to or with no HME process. The F2 physical mixture also had peaks much shorter than those of pure DQ (Figure 3B). From the comparison of Figure 3B with Figure 3C, it might be deduced that the portion of DQ peaks that was lost might have been shielded by other components of F2, namely, cholesterol and mono or diglycerides of medium-chain fatty acids (MCM) (see Table 1), rather than being fused with these components. Nevertheless, these lipids did play a role in enabling DQ to enter an aqueous solution. Otherwise, it was impossible for DQ itself to get into the aqueous suspension used for IM injection. 

### 3.5. Duration of Protection Provided by Lipids-Based Decoquinate Formulation 

F2 had a 40% drug load, higher than F3 (20%), but compositions were the same. Thus, F2 was suspended in saline and administered to mice by IM to further determine how long decoquinate could be effectively released to protect mice from *Plasmodium* infection. F2 was placed in six groups of mice for days ranging from 30 to 180. SPZ inoculation was given to each group of the mice only one time at different time lengths a month apart. IVIS images collected from the third month to the sixth month are shown in Figure 4A. All mice inoculated from the first month to the fourth month were completely free of the signals showing *Plasmodium* infection, suggesting that F2 could protect all animals for up to 4 months.

However, the animals inoculated with SPZ 150 days and 180 days after the decoquinate depot showed, in spite of the weak signals, the infection (Figure 4A and Table 2), indicating that they had not been completely protected by the SRFD from *berghei* malaria. The longest duration for a decoquinate depot to have complete protection was estimated to be between 120 and 150 days. According to the report [23], the minimal inhibition concentration (MIC) of DQ in the blood is 5.12 ng/mL, the level that could be maintained before day 120 before the SPZ inoculation. The vehicle control composed of cholesterol and MCM (F8) was unprotective for mice against *Plasmodium* infection.

### 3.6. Sustained Decoquinate Concentrations in the Blood 

The blood specimens were collected by cardiac puncture from C57 mice. The experiments paralleled the one with data shown in Figure 4A. The blood was drawn on different days shown in Figure 4B and DQ was measured by the LC-MS system as described in the method section. Each point was from one animal and plotted as ng/mL against time (days). DQ concentrations in the blood showed that DQ reached the highest level on day 5 and then gradually declined from day 30 to day 180. The declining trend of the curve against time represented DQ released from the depot stored in the muscle into the blood, which might correlate with the chemoprophylactic efficacy. The efficacy of causal prophylaxis resulted from DQ released from the DQ depot in the muscle, more likely via the bloodstream, to other parts of the body such as the liver and the skin.

The mean particle sizes of HME-made SRFD were measured and shown to be between 31–52 µm. The SRFD (F2), used in the experiments for Figure 4A,B, had a mean particle size of 33 µm with 90% of the particles within 58 µm (Figure 4C).

### 3.7. The Optimal Dose of Decoquinate for Protection of Mice Exposed to Plasmodium Infection 

F2 at a dose of DQ 200 mg/kg depot in the muscle could protect mice from *berghei* malaria and the duration of the protection lasted 4 months. To determine the minimal effective dose, F2 was placed intramuscularly in mice at a dose of DQ 50, 100, 150, and 200 mg/kg. Mice were inoculated with the SPZ one month after the F2 injection. Parasitemia was examined 4 to 6 days after the SPZ inoculation. Mice with F2 placement at DQ 50 mg/kg and 100 mg/kg were shown to have parasitemia to some extent and were not sufficiently protected from the *Plasmodium* infection (Figure 5A). Survival rates were calculated 75 days after the inoculation. Mice dosed with DQ 50 mg/kg could not survive by day 30 compared to day 9 of vehicle mice. By day 75, mice dosed with DQ 150 mg/kg and 200 mg/kg had survival rates of 100%; the survival rate of mice with DQ 100 mg/kg was 66.7%. The results indicate that the effective DQ dose in F2 should be between 100 mg/kg and 150 mg/kg to completely protect mice from *Plasmodium berghei* infection (Figure 5B).

### 3.8. Toxicity of Lipids-Based Decoquinate Formulation

Due to a limited volume that could be given to mice intramuscularly, toxicity assessment was performed by giving mice F2 with a maximal dose of DQ 200 mg/kg. F2 was composed of cholesterol, MCM, and DQ (Table 1). The animals showed no signs of acute adverse effects. Solid sticks of F4 and F5 could only be given by SQ injection. Animals administered subcutaneously with the solid form of the SRFD at DQ 200 mg/kg seemed to have manifested immediate distress and some degree of moving difficulty after the injection. Animals appeared to be more tolerant when dosed with cholesterol-based SRFD or cholesterol–MCM-based SRFD in an aqueous suspension injected by IM.

## 4. Discussion

Intragastric administration of nanoparticle formulations of DQ to mice provided an excellent efficacy of causal prophylaxis targeting the intrahepatic stage by killing parasites before they enter the blood [20]. Only when DQ is made into nanoparticles and sufficiently absorbed [20,21] can DQ reach and act on the sites where parasites are. Alternatively, DQ may be made in sustained-release formulations suitable for parenteral administration, long-term storage, and slow release of API, which can provide causal prophylaxis of *Plasmodium* infection [23]. This approach may help those who often forget to take the prophylactic medications as prescribed by physicians. The creation of SRFD could uniquely provide long-term prophylaxis for malaria.

PLGA has been widely used for sustained drug delivery [24]. Formulations made of PLGA as a carrier provide effective treatment of various medical conditions such as psychiatric diseases [12]. However, PLGA and DQ may not be compatible in forming a formulation of releasing API sufficient for the chemoprophylaxis of malarial infection. DQ is practically water-insoluble and extremely lipophilic. It is not an easy task of finding an appropriate carrier for DQ in sustained-release formulation. Originally, cholesterol was used as a control excipient for the SRFD. Surprisingly, it became a major carrier of DQ in SRFD, which provided excellent protection for mice inoculated with *Plasmodium berghei* SPZ. Furthermore, the SRFD prepared by HME such as F1, F2, and F3 released DQ (Figure 1A) sufficient for prophylactic efficacy (Figure 1B,C), better than F9, which was prepared by the emulsion method. Formulations containing PLGA, whether prepared by HME (F4) or by dissolving the components in organic solvent NMP (F10), did not release sufficient DQ to fight against *Plasmodium* SPZ invasion (Figure 1 and Figure 2). Other polymers (PLA, PGA, PCL) were also assessed and proved to be unsuccessful in making an SRFD for the long-term prophylaxis of malaria (data not shown).

Mild conditions such as the range of temperatures set well below the melting points of each component of the SRFD were chosen in the HME process so that active as well as inactive ingredients could be kept intact without structural changes in the molecules. Under such HME settings, the drug particles of the preparations could be obtained in the micrometer range (Figure 3A) to meet the requirement of the sustained release of DQ. There have been reports that large drug particles as the intramuscular reservoir, not fine particles such as nanometer particles, favor long-lasting drug exposure [23,26,27,28]. Large particles derived from different drug compositions can lead to the delayed release of API to last for many days in vitro (Figure 1A) and for as long as 180 days in vivo (Figure 4B). In contrast, nanoparticles of DQ formulation prepared by HME had drug dissolution in a short period of time from 1 to 9 h [20].

The SRFD was optimized through the screening of various formulations as shown in Figure 1 and Figure 2. The HME method is highly efficient for large-scale preparation and may have improved product quality by enhancing the mixing of all components and controlling the temperature as well as the whole processing [29]. XRD analyses showed crystalline peaks of a small fraction of DQ from HME extrudates, suggesting that DQ had not been completely miscible with other formulation components. This might negatively impact the release of DQ and compromise long-lasting prophylaxis. However, compared to the XDR pattern of pure DQ (Figure 3A), there was still more than 80% of loaded DQ (Figure 3C) complexed with the formulation components, which ensured the sustained-release of DQ over a long-extended period and protected the infected mice. Further assessment, as shown in Figure 4 and Figure 5, demonstrated that the SRFD at a dose of DQ 150–200 mg/kg could release DQ from the depot formulation (F2) stored in the muscle over 120 days, which had been enough for mice protection from *Plasmodium* SPZ infection. The scenario is different from previous studies in that nanoparticle formulations of DQ were prepared for the oral route, which significantly improved the bioavailability and efficacy of DQ [20,21]. The effective dose of DQ by the oral route daily was much lower than that of the SRFD: only 3 to 5 mg/kg for causal prophylaxis of malaria in mice. For long-term prophylaxis of malaria, a relatively large dose of DQ was used to sustain the drug release for as long as possible.

The study conducted by Li et. al. used peanut oil as a vehicle to prepare a microparticle-sized depot formulation of DQ and high-pressure homogenization to reduce the particle size to less than 15 µM [20]. SRFDs prepared by HME in this study were not further processed by additional physical manipulation such as ultrasonication and high-pressure homogenization. The HME extrudates could be homogeneously suspended and easily handled in an aqueous solution. Partial protection (67%) (Figure 5B) for mice inoculated with *P. berghei* SPZ one month after SRFD placement was 100 mg/kg, which is equivalent to the effective dose of 120 mg/kg from the study of assessing oily suspended DQ for the long-term (8 weeks) chemoprophylaxis of malaria [23].

In pharmaceutical applications, cholesterol is commonly used in liposomal preparation [30,31]. Cholesterol was mixed well with DQ in constructing SRFDs via HME. It might also help release DQ from the muscle depot. The SRFD with cholesterol as an inactive ingredient prevented *Plasmodium* infection much better than expected. The use of cholesterol in the sustainable formulation of drug release has not been reported previously and it is the first time that it has been used for forming antimalarial agents, specifically DQ, in the depot formulation.

Cholesterol is a natural molecule that constitutes a structural component of about 30% of all animal cell membranes [32]. The molecule is essential to maintaining both the integrity and fluidity of the membrane structure. In the clinic, as compared to the cholesterol content of the HDL particles, abnormal cholesterol levels such as higher blood LDL, especially higher LDL particle concentrations and smaller LDL particle sizes, contribute to myocardial infarction (heart attack), stroke, and peripheral vascular disease. There has been very little correlation between the amount of cholesterol used in the drug formulation and an increased risk of cardiovascular disease. Given the fact that the pool size of cholesterol in the human body is so large and its metabolic routes are so diverse, the amount of cholesterol used should be safe. On the other hand, since cholesterol has the structure of the tetracyclic ring in a trans conformation, it makes all but the side chain of cholesterol rigid and planar. In this structural role, cholesterol also reduces the permeability of the plasma membrane to neutral solutes, hydrogen ions, and sodium ions. The nature of cholesterol might make it suitable and valuable for its role within the drug formulation to control the release of antimalarial agents as well as other therapeutic drugs. Compared to polymers commercially available, cholesterol is cheap and readily available.

The implant made by melt exclusion traditionally needs to be placed surgically or by using a large-diameter needle [33]. As for SRFD F3, it was injected in the forms of an aqueous suspension and a solid stick. The aqueous suspension could be handled more easily and provided a better outcome than the solid form. DQ was injected intramuscularly and resided in situ for slow and constant release. Cholesterol and medium-chain fatty acids could favor the release of DQ slowly but efficiently for the inhibition of *Plasmodium* parasites. These lipids could be absorbed by surrounding tissues faster than DQ, and then metabolized or excreted. Therefore, there was no need to remove any injected materials. The DQ of the SRFD residing in the muscle tissue could be diffused to local surrounding tissues or could enter the bloodstream and then spread to the sites where the killing event of *Plasmodium* parasites occurred. The target sites were the peripheral tissues, especially the liver and the skin. The dose of DQ ranging from 100 to 150 mg/kg in the SRFD was fully effective in preventing *Plasmodium* infection. To translate this dose of DQ for human beings, API 10 to 15 mg/kg or 1050 mg may be needed for injection.

Primaquine and tafenoquine are used as primary and terminal prophylaxes for malaria [34]. Potential dose-dependent acute hemolytic anemia in individuals with glucose-6-phosphate dehydrogenase deficiency (G6PDd), and often the unavailability of testing for G6PDd, limits their use [35]. The combination of atovaquone–proguanil has been used as a causal and suppressive prophylactic agent [36] to delay the emergency. These medications prescribed as a prophylactic agent of malaria require frequent administration. DQ has multistage antimalarial activity and might also be used as a causal prophylactic and suppressive agent but with little cross-resistance with atovaquone [37]. A DQ depot may provide an excellent alternative to current prophylactic remedies to overcome the incompliance of patients. DQ has been proven to be safe as a veterinary drug for many years [38]. No eminent adverse effects were found when the nanoparticle formulations were given to mice either by intragastric administration or by intravenous injection [21,22]. Adverse effects or toxic signs were not observed in mice when SRFD suspension was placed by IM. Here, the SRFD was produced by HME, a process that is readily scalable and highly efficient. Nevertheless, preclinical studies should not begin until the full toxicity results of decoquinate in the blood upon being dosed intramuscularly are obtained and understood and the maximum concentration without causing health complications in long-term administration has been determined. The model of large animals such as monkeys and dogs may be necessary to further assess the long-acting efficacy of DQ and the adverse effects of drug exposure in the blood or key tissues such as the liver and the skin.

## 5. Conclusions

Sustained-release formulations of DQ for the effective prophylaxis of malaria were prepared by using the hot melt extrusion method and cholesterol as a major drug carrier. An intramuscularly injected depot of decoquinate is one dose of administration for the long-term prophylaxis of malaria at the liver stage. It may play a role as a “vaccine”, but with a low cost and easy to produce and handle. The newly created formulation utilizes naturally available materials that can be metabolized or degraded or excreted by animals or humans with no need of removing the depot substances by surgery. Thus, decoquinate released slowly from the muscle depot was effective against liver stage malaria for a long period of time.

## 6. Patents

Intramuscular depot of decoquinate compositions and method of prophylaxis and treatment thereof. Inventors: Hongxing Wang, Shuanghong Liang, Yinzhou Fan, Xiaoyi Huang, Li Qin, and Xiaoping Chen. International Application No.: PCT/CN2018/087307; Patent No.: ZL 2018 8 0001391.5. Granted by Chinese Patent Office. Patent issue No.: CN 109496152B; Issue Date of Patent: 18 June 2021.

## Figures and Tables

**Figure 1 pharmaceutics-14-02813-f001:**
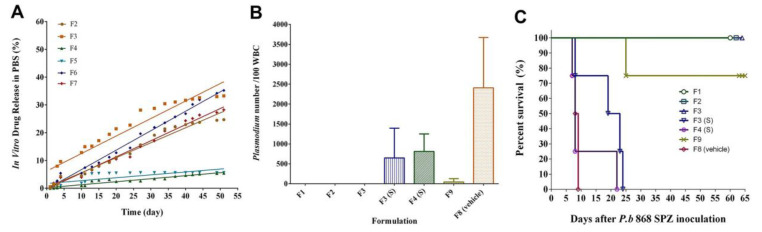
In vitro dissolution and in vivo efficacy assessment of sustained-release formulation of decoquinate (SRFD). (**A**) In vitro release of DQ from the solid form of SRFD. Each data point represents the percentage of cumulative drug released from the solid formulation of DQ, the drug released and accumulated from all previous times. (**B**) Microscopic examination of parasitemia in mice. The examination was carried out 4 days after the mice were infected by the *Plasmodium berghei* sporozoites. Lipids-based formulation of DQ was placed 34 days before *Plasmodium* infection. F1, F2, F3, and F8 were each suspended in saline and given to mice intramuscularly (IM). F4 was injected in solid form (stick) subcutaneously; F3 was also given in solid stick. F9 was prepared and injected as an emulsion. (**C**) Survival rates. The mice from the experiments shown in (**B**) were raised for 60 days and survival rates were counted 60 days post-inoculation.

**Figure 2 pharmaceutics-14-02813-f002:**
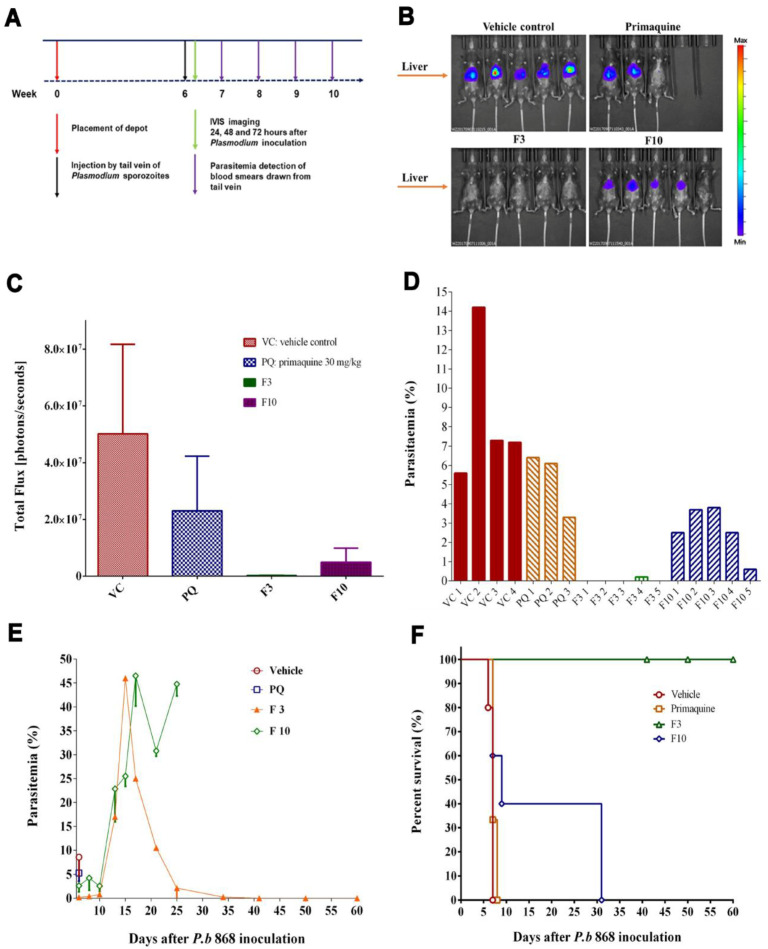
Evaluation of SRFD (F3) in malaria prophylaxis. (**A**) A schematic depiction of animal experimental design for the data shown in (**B**–**F**). (**B**) F3 at a dose of DQ 200 mg/kg, vehicle control of F3 (no DQ), F10 dissolved in NMP, and primaquine phosphate (PQ) at a dose of 30 mg/kg dissolved in saline, were given to mice by IM. Six weeks later, mice were inoculated with SPZ of *Plasmodium berghei* ANKA 868 expressing firefly luciferase via the tail vein. Images were acquired by using In Vivo Image Systems (IVIS) 46 h after the inoculation. (**C**) Counts presented as the flux of photons, corresponding to images in (**B**) from IVIS detection. (**D**) Parasitemia detection 6 days after *Plasmodium berghei* SPZ inoculation. (**E**) Parasitemia was detected periodically over 60 days after SPZ inoculation. The lines were interrupted at the time when animals were seriously ill and euthanized. Note that one animal in the F3 group was infected but recovered, parasitemia level peaked at day fifteen, then down to zero on day 35, and no recrudescence occurred. (**F**) The percentage of animals that survived 60 days after SPZ injection.

**Figure 3 pharmaceutics-14-02813-f003:**
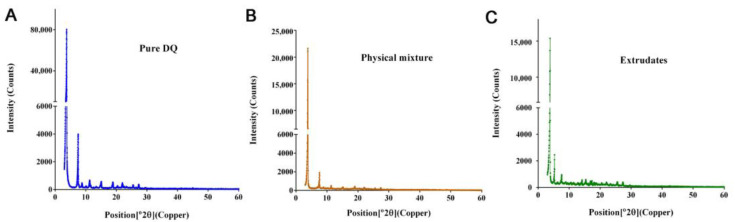
Physical features of SRFD. (**A**) X-ray diffraction graph of pure DQ powder; (**B**) X-ray diffraction graph of F2 made by HME; (**C**) X-ray diffraction graph of physical mixture of the same components of F2 prior to HME process. The peak heights of DQ from HME extrudates accounted for only a small percentage of the peaks of pure DQ.

**Figure 4 pharmaceutics-14-02813-f004:**
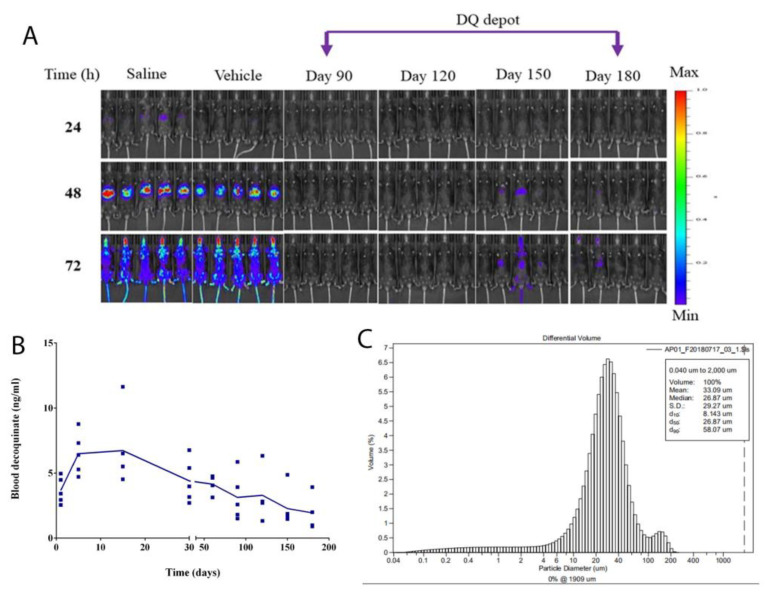
Duration of malaria prophylaxis provided by SRFD in mice with *Plasmodium* infection. HME extrudates (F2) were suspended in saline and given to separate groups of mice (C57) at a dose of DQ 200 mg/kg by IM. After SRFD replacement, mice were, respectively, inoculated with *Plasmodium berghei* SPZ at different time lengths. New mice for saline and vehicle controls were replaced and matched with experimental mice in each group. (**A**) Images shown are from experimental groups with the SPZ inoculation on days 90, 120, 150, and 180. Images from the early time points before 90 days are not shown. Mice were monitored by IVIS 24, 48, and 72 h after SPZ injection. At 24 and 48 h, the signals if any were localized in the abdominal area (liver) and then spread to the whole body at 72 h or after 48 h. (**B**) Parallel experiments were performed in the same kind of animals (C57) as the mice in (**A**). The same SRFD (F2) was administered at a dose of DQ 200 mg/kg by IM to the mice. Blood samples were drawn at different days by cardiac puncture and the blood concentrations of DQ were measured by LC-MS. (**C**) Mean particle size of HME extrudates (F2).

**Figure 5 pharmaceutics-14-02813-f005:**
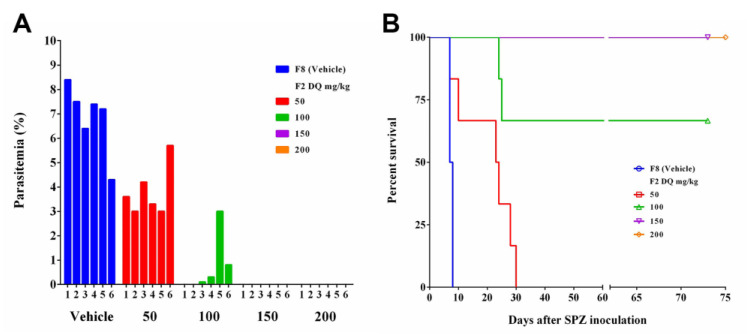
In vivo efficacy validation of effective dose of DQ in SRFD. SRFD (F2) with different doses of DQ were placed one month before *P. berghei* 868 SPZ inoculation. (**A**) Parasitemia was detected by examining blood smears from mice 6 days after the SPZ inoculation. (**B**) Percent survival rates of animals 75 days after the SPZ inoculation.

**Table 1 pharmaceutics-14-02813-t001:** Compositions of formulations.

Preparation of DQ Depot Formulation (F1–F8) by HME						
Formulation	Chol (g)	MCM (g)	PLGA75/25 (g)	P188 (g)	DQ (g)	Drug Load (%)
F1	8.40		--		3.6	30.00
F2	7.50	1.50	--		6.00	40.00
F3	10.50	1.50			3.00	20.00
F4	7.50	1.50	1.50		4.50	30.00
F5	7.50	0.75	2.25		4.50	30.00
F6	9.00	1.50			4.50	30.00
F7	9.00			1.50	4.50	30.00
F8	10.80	1.20				
F9	75 (mg)				100 (mg)	57.00
F10			115 (mg)		115 (mg)	50.00

Note: F9 and F10 were not prepared by the HME method. F9 was prepared by the emulsion method by suspending DQ in triglyceride. F10 was prepared by the solvency method by suspending DQ in NMP. Abbreviations: Chol: cholesterol; MCM: Capmul MCM 8; DQ: decoquinate. PLGA 75/25: L-lactide-co-glycolide with L/G ratio of 75/25. NMP: N-methyl-2-pyrrolidone.

**Table 2 pharmaceutics-14-02813-t002:** Chemoprophylaxis of DQ in C57 black mice infected with *Plasmodium berghei* (*P.b.*).

Days for Inoculation	Saline			Auxiliary Control			Formulation 3 (F3)		
	Mice (*n*)	Infected	Survived	Mice (*n*)	Infected	Survived	Mice (*n*)	Infected	Survived
day 30	5	5	0	5	5	0	5	0	5
day 60	5	5	0	5	5	0	5	0	5
day 90	5	5	0	5	5	0	5	0	5
day 120	5	5	0	5	5	0	5	0	5
day 150	5	5	0	5	5	0	5	0	5
day 180	5	5	0	5	5	0	5	0	5
day 210	5	5	0	5	5	0	5	0	5
day 240	4	4	0	4	4	0	5	1	5
day 360	4	4	0	4	4	0	5	1	4

Note: F2 suspended in saline, vehicle (F8) suspended in saline, and saline alone, given to mice by IM. Mice in each treatment group were then divided into subgroups for inoculation of *P.b*. SPZ. The inoculation was arranged according to the duration of drug placement (see the table). Parasitemia rates and survival rates were summarized 6 days and 30 days, respectively, after the inoculation.

## Data Availability

No data reported.

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
