# Peer review of "Preliminary Assessment of Intramuscular Depot of Lipid-Based Decoquinate Formulation for Long-Term Chemoprophylaxis of Malaria"

_pharmaceutics, 2022, doi:10.3390/pharmaceutics14122813_

Round 1
Reviewer 1 Report
The manuscript presented for review describes the results of research on the formation of the depot system for subcutaneous implantation to obtain the long-term release of Decoquinate, a drug according to the authors having long-term chemoprophylaxis of Malaria. This work is a continuation of previous authors' articles on the formulation of Decoquinate-releasing nanosystems (drug nanoparticles, drug transport through liposomes). Practically the same system containing liposomes and Decoquinate but dispersed in the PLAGA matrix was described in the reviewed work.
In general, the level of the work carried out in terms of its content and professionalism should be considered high. After analyzing the results, however, in my opinion, the results of a few studies that should be done to clarify many of the problems I have encountered are missing. The content of the manuscript should be expanded, it is necessary to clarify the issues that arise while studying the presented results. Below is a list of my concerns, to which I expect a full answer. I believe that the work may be accepted for publication only if the authors dispel these doubts.
- Abstract;
this section should be thoroughly redrafted, very little is known about the structure of the presented release system, no clear summary of what the proposed system is better than the periodic administration of the drug used or the Decoquinate release systems described earlier.
- Introduction;
there are no examples of the use of antimalarial drug-eluting systems (e.g. Antimicrob Agents. Ch. 2016, 60, 1304–1318., Biomaterials 2014, 13, 7940–7950., Parasites Vectors 10, 117 (2017) .https: // doi. org / 10.1186 / s13071-017-2018-7 and many others, including those on the market)
how do we know that Decoquinate has a corresponding anti-Plasmodium effect ??? The current state of knowledge is not convincing - I have not found clinical or advanced preclinical evidence as regards the effectiveness of such therapy in mammals, especially humans. Why did the authors not use this system for similar drugs with clinically proven effects?
Why do the authors not cite and then not relate their results to those presented in previous publications on a very similar topic; nanoparticle formation [Nanomedicine: Nanotechnology, Biology, and Medicine Volume 10, Issue 1, January 2014, Pages 57-65.] and the most recent one on liposome formation [Decoquinate liposomes: highly effective clearance of Plasmodium parasites causing severe malaria. Malar J 21, 24 (2022). https://doi.org/10.1186/s12936-022-04042-8]. Is the innovation about the previous articles just the placement of the liposomes described above in the PLAGA matrix?
- Figure 1
For clarity, the authors should include a graph also showing the distribution of daily doses of the released drug over time. There seems to be a strong burst effect. The presented drug release process is ineffective at best, after 55 days we can observe that only a maximum of about 40% of the total administered drug has been released. So, what happens to the drug then? It will be released during the second phase of degradation of the polymer matrix. This process can be very violent, resulting in very high daily doses in the late period of therapy. Please address this issue in the manuscript.
By no means is the rate of release shown here not linear, is not proportional to time, as suggested by the straight lines in Figure 1A. I do not understand what is shown in Figure 1B. Please change the description for this drawing and simplify it to make it legible.
- Rozdział 3.2
What are the average daily doses obtained? The daily dose recommended earlier in the publication on the action of Decoquinate orally administered dose is much lower (10 mg/kg). So, inferring from this, the drug dose in blood should not exceed about 1 mg/kg. Are high concentrations of drugs in blood immediately after implantation, which are noted, really nontoxic, especially for the liver??
-Chapter 3.5, Figure 4
Are the drug levels during the first days non-toxic?? Probably even higher in the liver. The clear burst effect at high doses may be dangerous to the patient's health. Please comment on this appropriately in the manuscript.
- Chapter 3.7
To determine the usefulness of the treatment methodology used by the authors and to confirm the correctness of its choice for further studies, it is necessary to supplement the results with basic histopathological examinations (changes in liver tissue, heart tissue, and blood count) of the examined mice. Only then can it be stated whether there is any chronic toxicity of the organism causing possible organ changes during the therapy, which with high probability could have occurred.
- Disscusion, line 585
According to the presented calculation of the minimum amount of the drug in the presented system (1050 mg, for a patient 70 kg), the final described depot for an average person should have a minimum weight of over 3 g. Isn't that too large an intramuscular implant? Its degradation will take at least a year, and the risk of complications in the form of chronic inflammation or sudden drug emission in the late phase of implant degradation is rather serious. Please raise this issue in the discussion, how to counteract this threat?
- Discussion, line 610
The proposed preclinical studies are of course necessary. However, such studies are only useful when the full toxicity results of this drug in the blood are known, and its maximum concentration not causing health complications in long-term administration is determined.
Conclusion
The presented conclusions practically do not add anything. Is the presented system better than the previously described methods of administering this drug? If so, what is better than the system described by the authors in a paper published in January this year, which consists in administering liposomes with the drug directly intravenously?
Demonstrate advantages and disadvantages. Is it worth investing in further research on the system described in the reviewed work?
Author Response
Response to Reviewer 1 Comments
Point 1:
- Abstract;
this section should be thoroughly redrafted, very little is known about the structure of the presented release system, no clear summary of what the proposed system is better than the periodic administration of the drug used or the Decoquinate release systems described earlier.
Response 1: The manuscript was originally written for a different journal, more orientated to the non-pharmaceutical journal. Now for “Pharmaceutics, the abstract has been thoroughly redrafted and much more fit this journal. We would like to thank the Reviewer.
Point 2:
- Introduction;
there are no examples of the use of antimalarial drug-eluting systems (e.g. Antimicrob Agents. Ch. 2016, 60, 1304–1318., Biomaterials 2014, 13, 7940–7950., Parasites Vectors 10, 117 (2017) .https: // doi. org / 10.1186 / s13071-017-2018-7 and many others, including those on the market)
Response 2:
Thank the reviewer, the author added these references as 6,7, 8. We would like to thank the Reviewer.
As far as we know, there are no antimalaria drugs currently on the market. So all studies are still preclinical.
Point 3: how do we know that Decoquinate has a corresponding anti-Plasmodium effect ??? The current state of knowledge is not convincing - I have not found clinical or advanced preclinical evidence as regards the effectiveness of such therapy in mammals, especially humans. Why did the authors not use this system for similar drugs with clinically proven effects?
Response 3 Please note, the references are new list in the revised manuscript
Earlier studies by others found that DQ had low nanomolar (2.6 nm) activity against in vitro liver stage Plasmodium parasites, about 3000-fold lower than that of primaquine [15-17] and a single 5 mg/kg oral dose completely protected mice against Plasmodium infection caused by mosquito bite [16]. Interestingly, among all compounds assessed, DQ had the highest therapeutic index (>2500) for treating P. falciparum 3D7 strain [17]. DQ is a cytochrome bc1 inhibitor with little cross-resistance against a panel of 5 strains of P. falciparum parasites resistant to the antimalarial drug atovaquone, another cytochrome bc1 inhibitor [17]. These important findings are scientific bases for developing this molecule as a useful antimalarial drug. Since then, however, work has been seldom followed up to advance this drug development. The most likely reason is that without modification or formulation to improve the solubility, pure DQ is extremely difficult to use in in vivo studies.
Point 4: Why do the authors not cite and then not relate their results to those presented in previous publications on a very similar topic; nanoparticle formation [Nanomedicine: Nanotechnology, Biology, and Medicine Volume 10, Issue 1, January 2014, Pages 57-65.] and the most recent one on liposome formation [Decoquinate liposomes: highly effective clearance of Plasmodium parasites causing severe malaria. Malar J 21, 24 (2022). https://doi.org/10.1186/s12936-022-04042-8]. Is the innovation about the previous articles just the placement of the liposomes described above in the PLAGA matrix?
Response 4 Please note, the references are new list in the revised manuscript
In the Introduction, we describe our previous work relevant to this paper:
As an anti-coccidiostat used for years, DQ has an excellent safety profile [18]. DQ can effectively inhibit chloroquine-sensitive P. falciparum as well as chloroquine- or multidrug-resistant P. falciparum in infected human erythrocytes and protect mice from severe malaria [19, 21]. To solve the problem of water insolubility and to improve bioavailability, DQ has been made into nanoparticles without compromising its antimalarial potency as an oral dosage form targeting liver stage Plasmodium infection. The nanoparticle formulations of DQ are very potent at inhibiting Plasmodium berghei in vitro at the liver stage (IC50<0.5 nM) and highly efficacious (1~3 mg/kg) in providing protection of mice from liver stage malaria [20, 21]. The in vivo efficacy of DQ against liver stage Plasmodium might be also associated with its pharmacokinetic feature in that there is high enrichment of DQ in the liver whether the nanoparticles were orally or intravenously administered [22]. Recently, Li, et al have created formulations of DQ in oil-based carriers to provide extended efficacy through continuous drug release over a prolonged period from a single injection of the depot to effectively protect mice from Plasmodium infection [23].
Point 5:
- Figure 1
For clarity, the authors should include a graph also showing the distribution of daily doses of the released drug over time. There seems to be a strong burst effect. The presented drug release process is ineffective at best, after 55 days we can observe that only a maximum of about 40% of the total administered drug has been released. So, what happens to the drug then? It will be released during the second phase of degradation of the polymer matrix. This process can be very violent, resulting in very high daily doses in the late period of therapy. Please address this issue in the manuscript.
Response 5
The formulation evaluated was sustained release formulation (see revised abstract). The purpose is to prolong the activity of the drug so that one dose is given and the dose is as large as possible. Ideally, the drug release was slow and lasts for a long period of time. So there may be no violent and burst release of the drug. It is important to show also the distribution of daily doses of the released drug over time. However, daily doses might cause more pain for animals, possibly not by chemical toxicity but by the pain caused by the physical stimulation of large volumes. The drug as veterinary medicine caused very few adverse effects (37). According to available literature, the majority of this drug is unlikely to degrade in animals or humans. The most likely fate of decoquinate is to be excreted through the gut (37).
Indeed, “after 55 days we can observe that only a maximum of about 40% of the total administered drug has been released”. It was only observed for 55 days. If it was observed for 180 days or longer, the drug release would be 100%. In fact, it explains why our formulation can protect animals for as long as 120 days, not only 55 days.
The revised abstract describes that “One-time administration of pharmaceutical agents in such a slow-release system may serve patients with no concerns about compliance.”
Point 6:
By no means is the rate of release shown here not linear, is not proportional to time, as suggested by the straight lines in Figure 1A. I do not understand what is shown in Figure 1B. Please change the description for this drawing and simplify it to make it legible.
Response 6
The figure legends for Figures 1A, 1 B, and 1C were rewritten to improve legibility.
Point 7:
What are the average daily doses obtained? The daily dose recommended earlier in the publication on the action of Decoquinate orally administered dose is much lower (10 mg/kg). So, inferring from this, the drug dose in blood should not exceed about 1 mg/kg. Are high concentrations of drugs in blood immediately after implantation, which are noted, really nontoxic, especially for the liver??
Response 7
The difference between the current manuscript from the previous publication is that in the previous experiment, the drug was administered orally and daily; the current study uses sustained release formulation and gives only a one-time drug dose intramuscularly. It was shown that 2000 mg/kg given to mice orally was nontoxic. “So, inferring from this, the drug dose in blood should not exceed about 1 mg/kg.” It has been indicated that the oral dose of HME DQ that was 650 folds greater than the efficacious in vivo dose did not result in any prominent adverse effects in mice. In this study, no observable toxicity of 200 mg/kg administered intramuscularly was seen. Indeed, the livers should have been checked. However, they were only checked anatomically not histochemically, and no obvious toxicity was found. In future studies, liver toxicity should certainly be done.
Point 8:
Are the drug levels during the first days non-toxic?? Probably even higher in the liver. The clear burst effect at high doses may be dangerous to the patient's health. Please comment on this appropriately in the manuscript.
Response 8
Earlier studies by others found that DQ had low nanomolar (2.6 nm) activity against in vitro liver stage Plasmodium parasites, about 3000-fold lower than that of primaquine [15-17] and a single 5 mg/kg oral dose completely protected mice against Plasmodium infection caused by mosquito bite [16]. Interestingly, among all compounds assessed, DQ had the highest therapeutic index (>2500) for treating P. falciparum 3D7 strain [17].
The drug levels during the first days did not show any toxic effects. The experiments were repeated many times and all mice are OK. The drug has high distribution in the liver whether it is given orally or intravenously. But the previous study did not show any liver toxicity (22)
Point 9:
To determine the usefulness of the treatment methodology used by the authors and to confirm the correctness of its choice for further studies, it is necessary to supplement the results with basic histopathological examinations (changes in liver tissue, heart tissue, and blood count) of the examined mice. Only then can it be stated whether there is any chronic toxicity of the organism causing possible organ changes during the therapy, which with high probability could have occurred.
Response 9
Totally agree. In future studies, small and large animals should be examined to see if there is any chronic toxicity. For now, we did not perform any histopathological experiments and have no supplementary results to submit.
Point 10
- Disscusion, line 585
According to the presented calculation of the minimum amount of the drug in the presented system (1050 mg, for a patient 70 kg), the final described depot for an average person should have a minimum weight of over 3 g. Isn't that too large an intramuscular implant? Its degradation will take at least a year, and the risk of complications in the form of chronic inflammation or sudden drug emission in the late phase of implant degradation is rather serious. Please raise this issue in the discussion, how to counteract this threat?
Response 10
The half-life of decoquinate is about 4 ½ hours once it is released into the blood (19). Supposedly it takes 6 months to release all the drug from the depot in the muscle, it will be excreted through the gut (37). It is calculated that for humans, the dose would be 10 times less than for mice. Therefore, For human beings, 150mg/kg, to 15 mg/kg, 15 X 70 (kg), the one-time dose would be 1050 mg. This can be given in two or three injections to reduce discomfort. It would be practically feasible.
Point 11
- Discussion, line 610
The proposed preclinical studies are of course necessary. However, such studies are only useful when the full toxicity results of this drug in the blood are known, and its maximum concentration not causing health complications in long-term administration is determined.
Response 11
Good point. I will add this issue to the discussion.
Point 12
Conclusion
The presented conclusions practically do not add anything. Is the presented system better than the previously described methods of administering this drug? If so, what is better than the system described by the authors in a paper published in January this year, which consists in administering liposomes with the drug directly intravenously?
Demonstrate advantages and disadvantages. Is it worth investing in further research on the system described in the reviewed work?
Response 12
The published in January and this manuscript are totally different drug delivery systems and have completely different purposes. The liposome is intravenously given for the treatment of severe malaria, which is acute, life-threatening, and blood stage. It may need several doses. Nano-sized liposomes do not work as a delayed release of decoquinate (23). The depot is intramuscularly one dose, long-term, for prophylaxis, and liver stage. People who work on Malaria can understand more easily.
Sustained release formulation could play a role as a “vaccine”, but with lower cost and less trouble producing and handling than the vaccine. Walter Reed Institute in MD, US, is interested in making chemoprophylactic formulations for malaria (23). All of these points are well taken and will be added to the manuscript.

Reviewer 2 Report
This manuscript describes researches related with formulations of Decoquinate based on lipids. Using cholesterol as a carrier by thermal extrusion, this work found a good efficiency against Malaria as a long-term prophylaxis. This subject is very interesting since there are not useful vaccines against this disease and the prophylaxis is difficult especially in developing countries. The manuscript is well organized and written. However, some aspects should be improved. Therefore, I think that this manuscript should be corrected with major revisions, before being accepted for publishing in this journal, addressing the following comments:
Line 82: a definition of this abbreviation would be welcome.
Table 1: It is not clear if F2 was prepared by HME, because in some place of the text is described as a physical mixture. Which method was used for preparation of F9 and F10? Physical mixture? A better explanation is these footnote and in the main text should be appreciated.
Figure 1a: The results of the physical mixture of F2 and F9 and F10 are not shown. The addition of these values would be useful and additional explanations related with this point should be included.
Fig. 1: Was the same preparation used in fig 1b also for the release studies in fig. 1a? Probably the dissolution rate will be different suspended in saline solution that in solid stick. Please a better explanation would be appreciated.
Fig, 1a: The maximal release amount is lower than 30% in most of the cases. In this aspect, most of the drug is not used and should be metabolized in the body. A certain explanation and discussion related with this point would be appreciated.
Fig. 3: A XRD profile of the formulations without DQ would be appreciated. It is not clear if the peak at around 4º corresponds to excipients, cholesterol or others, or to DQ.
Fig. 3c: The physical mixture of F2 is described. However, this material is not the same of described in Fig 1 dissolution rate, possibly prepared by HME. This is confused and a higher coherency is necessary. A better explanation in the caption of figure or in the text would be welcome.
Lines 485: grammar and style revision.
Lines 496-497: No comparison between the release of DQ of this work and those data of previous works with nanoparticles. This comparison would be useful for discussion and would be welcome.
Lines 515-517: Why these data are not shown? An explanation and comparison results would be appreciated.
Lines 529-536: I understand the optimism and hope of authors, however the reality is more complex. The crystallinity of physical mixture is similar that the extruded formulation (Fig 3). A zoom out of these diffractograms will show this. This explain the low release percentage of the formulations. I suggest to authors the comparison with other excipients that can amorphize the DQ, other gels with surfactants or carriers as clay minerals.
Other suggestion to authors is to prepare the formulations without HME, solving DQ in an organic solvent and prepare the mixture with lipids with this organic solution eliminating after the solvent. This material will show a different XRD profile and release rate and amount that the samples of this work and the comparison can be useful and would increase the interest of the manuscript.
Lines 539-542: A further comparison of the Li et al, work with peanut oil with these data would be appreciated, especially in the release studies.
Conclusions: Too short.
References: An homogenous format would be appreciated.
Author Response
Response to Reviewer 2 Comments
The authors thank the reviewer so much for the comments and suggestions, which help us improve the manuscript tremendously.
Point 1:
Line 82: a definition of this abbreviation would be welcome.
Response 1:
The definition was added. Thank you.
Point 2:
Table 1: It is not clear if F2 was prepared by HME, because in some place of the text is described as a physical mixture. Which method was used for preparation of F9 and F10? Physical mixture? A better explanation is these footnote and in the main text should be appreciated.
Response 2:
The preparation methods for F9 and F10 are commonly used for most poorly soluble drugs. In this study, they were used as a comparison. F2 was prepared by HME. The same components of F2 were prepared by the HME process and by simple mixing (physical mixture) as a control in XRD analysis. The figure legend and descriptions in the Method and Results were made clear in the revised version. For example, in the figure legends, it is stated that “(C) X-ray diffraction graph of physical mixture of the same components of F2 (no HME process).”
Point 3:
Figure 1a: The results of the physical mixture of F2 and F9 and F10 are not shown. The addition of these values would be useful and additional explanations related with this point should be included.
Response 3:
The physical mixture of F2 is not in the solid form. Simple mixing cannot make a solid. F9 is made as an emulsion, so it is not solid. F10 is made in dissolving the drug in NMP. The liquid form is not as easy as the solid form for in vitro drug dissolution tests. We tried to evaporate the solvent and use the solid to test the drug release, but no drug release can be detected.
Point 4:
Fig. 1: Was the same preparation used in fig 1b also for the release studies in fig. 1a? Probably the dissolution rate will be different suspended in saline solution that in solid stick. Please a better explanation would be appreciated.
Response 4:
The drug release test was performed in phosphate buffered saline (PBS, PH 7.4). The liquid form for administration by intramuscular injection was made by suspending the HME extrudates in normal saline with pH 5.5. Now thanks to the Reviewer, we make it clear to readers.
Point 5:
Fig, 1a: The maximal release amount is lower than 30% in most of the cases. In this aspect, most of the drug is not used and should be metabolized in the body. A certain explanation and discussion related with this point would be appreciated.
Response 5:
The tests were originally done only for the period of efficacy observation (34 days), So it was only observed for 55 days. If it was observed for 180 days or longer, the drug release would be 100%. In fact, it explains why our formulation can protect animals for as long as 120 days, not only 55 days.
Point 6:
Fig. 3: A XRD profile of the formulations without DQ would be appreciated. It is not clear if the peak at around 4º corresponds to excipients, cholesterol or others, or to DQ.
Response 6:
Agree with the Reviewer. The formulation without DQ should have been included in the XRD analysis. We will consider doing this if there is a chance. Right now, it is very difficult to do anything right now. Guangzhou is in serious condition with Covid 19 pandemic. The labs are closed. But we can assume that the height of the peaks is much shorter in the physical mixture and the HME formulation than the height of pure DQ.
Point 7:
Fig. 3c: The physical mixture of F2 is described. However, this material is not the same of described in Fig 1 dissolution rate, possibly prepared by HME. This is confused and a higher coherency is necessary. A better explanation in the caption of figure or in the text would be welcome.
Response 7:
As described in Response 2, the physical mixture of F2 is now changed to “(C) X-ray diffraction graph of physical mixture of the same components of F2 prior to HME process. In the Results, it is revised to “F2 physical mixture had the same formulation components of F2 prior to or no HME process. The F2 physical mixture also had peaks much shorter than those of pure DQ (Fig. 3C).
Point 8:
Lines 485: grammar and style revision.
Response 8:
The grammar and style have been checked and modified.
Point 9:
Lines 496-497: No comparison between the release of DQ of this work and those data of previous works with nanoparticles. This comparison would be useful for discussion and would be welcome.
Response 9:
Yes, it should be compared. The drug released is added to the discussion.
The scenario is different from previous studies in that nanoparticle formulations of DQ were prepared for the oral route, which significantly improved the bioavailability and efficacy of DQ [20, 21]. The effective dose of DQ by the oral route daily was much lower than that of SRFD, only 3 to 5 mg/kg for causal prophylaxis of malaria in mice. For long-term prophylaxis of malaria, a relatively large dose of DQ used was to sustain the drug release for as long as possible.
Point 10:
Lines 515-517: Why these data are not shown? An explanation and comparison results would be appreciated.
Response 10:
It is embarrassing to say that we had not kept those initial data because those polymers-containing formulations failed to protect mice. The mice were all infected. The pictures were ugly, and experiments terminated right afterward. The staff had no patience to carry out such a long period of the experiment.
Point 11:
Lines 529-536: I understand the optimism and hope of authors, however the reality is more complex. The crystallinity of physical mixture is similar that the extruded formulation (Fig 3). A zoom out of these diffractograms will show this. This explain the low release percentage of the formulations. I suggest to authors the comparison with other excipients that can amorphize the DQ, other gels with surfactants or carriers as clay minerals.
Response 11:
The following part is added to the DISCUSSION. “Large particles derived from different drug compositions can lead to the delayed release of API to last for many days in vitro (Fig. 1A) and for as long as 180 days in vivo (Fig. 4B). In contrast, nanoparticles of DQ formulation had drug dissolution in a short period of time from 1 to 9 hours [20, 21].”
Your suggestion of comparing with other excipients is well taken for future study to expand the options to make the slow-release drug system.
Point 12:
Other suggestion to authors is to prepare the formulations without HME, solving DQ in an organic solvent and prepare the mixture with lipids with this organic solution eliminating after the solvent. This material will show a different XRD profile and release rate and amount that the samples of this work and the comparison can be useful and would increase the interest of the manuscript.
Response 12:
Yes, your suggestions are excellent and all well accepted. Now it is a difficult time, and we have a deadline to send back the revision.
Point 13:
Lines 539-542: A further comparison of the Li et al, work with peanut oil with these data would be appreciated, especially in the release studies.
Response 13:
Li et al used peanut oil to make DQ depot emulsion which is a milk-like liquid. We did our drug release test using solid preparation. Maybe we can think of different ways to do the test.
Point 14:
Conclusions: Too short.
Response 14:
It is modified so that it gives more views.
Point 15:
References: An homogenous format would be appreciated.
Response 15:
Yes, thank you. I am trying to do it.

Round 2
Reviewer 1 Report
I received a fairly comprehensive assessment for most of the problems and comments, which I sent to the authors. Changes in the manuscript's text were not marked, which required additional unnecessary work. Amendments have been introduced in the manuscript that did not completely satisfy me. I still have some reservations, which I presented below.
Response 3.
I understand that the initial tests presented in the cited publications suggest the potential high effectiveness of the drug used in the studied system. However, I meant the fact that there is no confirmation of the point of this drug obtained based on clinical trials. This finally means that the purposefulness of the conducted research is quite uncertain after all. So please explain in the text of the manuscript why, despite the lack of adequate certification to allow use in planned treatment or prevention of malaria, this drug was chosen for conducted tests. Is it possible to use another similar drug with poor solubility, but with clinically confirmed effectiveness?
Response 5.
The presented description under the drawings is too complicated and too long. The explanations presented are valuable, but they should be brought into the content of the manuscript. The signature under the figures itself should be simple. This applies to all drawings posted.
Why for comparison was it not presented results of figure 1A (results obtained in vitro) parallel to the results pictured in figure 4 ( drug concentration in blood in vivo)? Why are both in vitro and in vivo releases completely different? In the data presented from the study of changes in the drug concentration in the blood (Fig. 4), depending on the time after the implantation of the depot, the statistical data should be processed and introduced to the chart (average standard deviation, confidence interval). Only after taking into account the data obtained from statistical calculations the course of the curve can be plotted.
Response 11.
So we agree that the results obtained are only the results of preliminary tests. Such information should appear in the title or/and in the manuscript abstract.
Author Response
The authors would like to thank the Reviewers for providing critics, comments and suggestions which help improve the quality of the manuscript. This time all revisions on the manuscript have been marked for your convenience.
To Reviewer 1:
Comments and Suggestions for Authors
I received a fairly comprehensive assessment for most of the problems and comments, which I sent to the authors. Changes in the manuscript's text were not marked, which required additional unnecessary work. Amendments have been introduced in the manuscript that did not completely satisfy me. I still have some reservations, which I presented below.
Point 1 (Response 3).
I understand that the initial tests presented in the cited publications suggest the potential high effectiveness of the drug used in the studied system. However, I meant the fact that there is no confirmation of the point of this drug obtained based on clinical trials. This finally means that the purposefulness of the conducted research is quite uncertain after all. So please explain in the text of the manuscript why, despite the lack of adequate certification to allow use in planned treatment or prevention of malaria, this drug was chosen for conducted tests. Is it possible to use another similar drug with poor solubility, but with clinically confirmed effectiveness?
Response 1:
Yes, we agree with you. This molecule has no certification as a commercially available drug for malaria yet. This molecule has liver-stage activity and is extremely hydrophobic, safe, and highly efficacious. All these characteristics are the rationale for us to carry out experiments.
Although we think it is an excellent strategy to do such a work, in the conclusion, we are quite conservative “The results may provide useful information, not only for preparing a formulation of long-acting decoquinate but also for developing a potentially controlled drug release system. One-time administration of pharmaceutical agents in such a slow-release system may serve patients with no concerns about compliance.”
What is added to the manuscript to meet your request:
“The in vivo efficacy of DQ against liver stage Plasmodium might be also associated with its pharmacokinetic feature in that there is high enrichment of DQ in the liver whether the nanoparticles were orally or intravenously administered (22). “ (Why we did work for DQ? The yellow mark is the reason)
“The lipophilic property may make DQ an appropriate candidate as a sustained-release formulation for the chemoprophylaxis of malaria. Recently, Li, et al have created formulations of DQ in oil-based carriers to provide extended efficacy through continuous drug release over a prolonged period from a single injection of the depot to effectively protect mice from Plasmodium infection [23]. Interestingly, the microparticle formulation provided a 2.2-fold longer drug exposure and 3-4 times longer prophylactic effect than the nanoparticle formulation.” (For there is no clinically confirmed effectiveness). I wrote “Although there have been experiments performed in animals to prove that DQ has excellent antimalarial activity, it has not become a candidate for clinical trials to evaluate the safety and efficacy of preventing or treating malaria. Long-acting injectable atovaquone nanomedicines for malaria prophylaxis have also been reported [24] but nanoparticles may not be as sustainable as microparticles for drug release. “
It seems that only the molecules that have liver-stage activity and are hydrophobic are suitable for making long-acting formulations. Antimalarials that are blood-stage, and water-soluble are not. There are only very limited antimalarial molecules that can be made as long-acting formulations. But we rather not say this in the introduction because we did not test every and each malarial drug and have no evidence to make statements. Atovaquone (ATQ) has the same mechanism of action as DQ. More importantly, there is no cross-resistance between the two. Therefore, we added this paper citation (24). But the formulation approach, the duration of protection, and the authors indicated protection can translate to humans at clinically achievable and safe drug concentrations, potentially offering protection for at least 1 month after a single administration. Our formulation can provide at least 4 months after a single administration.
It is our interest and belief that DQ is better than ATQ for long-acting formulations.
Point 2 (Response 5).
The presented description under the drawings is too complicated and too long. The explanations presented are valuable, but they should be brought into the content of the manuscript. The signature under the figures itself should be simple. This applies to all drawings posted.
Response 2:
According to your critics, I have checked through all figure legends and made them as simple and as clearas possible. Some unnecessary words have been removed (see each figure legend).
It is common to describe the drawings for various readers even if it appears to be “too complicated and too long”. Readers may be experts in malaria, formulation or in public health, and so on. Some readers including different reviewers look for more explanation of the figure caption “A better explanation in the caption of figure or in the text would be welcome.”
More complicated and long descriptions are the following:
Here are some examples from reference 24:
Example 1: Fig. 3 Efficacy testing of ATQSDN7. a Experimental scheme. Mice dosed on day 0 with placebo, oral atovaquone or intramuscular nanoparticle atovaquone formulation were challenged once with intravenous P. berghei sporozoites at a given interval after dosing (28 days is depicted). For 42 days after challenge, blood samples were obtained and monitored for parasitemia (dots). b Prophylactic efficacy of intramuscular ATQSDN7. Cohorts of mice treated on day 0 with indicated doses of ATQSDN7 were challenged at a given interval after dosing (red arrows). Black lines, dose-to-challenge interval; grey lines, 42 days follow-up monitoring period. Prophylaxis was successful (circles) if, in two independent experiments (each with a cohort of 3–5 mice) all animals remained parasite-free for 42 days after challenge. Prophylaxis failed (x) if patent parasitemia was detected in any mouse in the cohort. For failed regimens the actual intervals between challenge and failure are not depicted. Not shown, all concurrent placebo recipients developed parasitemia, and all concurrent oral atovaquone controls failed challenge on or before 7 days after dosing. In all comparisons, successful prophylaxis with intramuscular ATQSDN7 was superior to no-drug control at p ≤ 0.003 (Fisher’s exact test).
Example 2: Fig. 4 Causal vs suppressive prophylactic antimalarial activity. On day 0 mice were injected with 36 mg kg−1 ATQSDN7, then four paired cohorts (three mice per cohort) were challenged at the indicated times with 5000 P. berghei sporozoites (upward black arrows) or 150,000 infected erythrocytes (upward red arrows). In keeping with the 48 h duration of P. berghei ANKA liver stages in C57BL/6 mice, the challenges for each pair were staggered by 48 h. For all four pairs, the sporozoite-challenged mice (green bar with red arrowhead, to indicate liver then possible erythrocytic phases) remained parasite-free at 42 days after challenge (hatched). All blood stage-challenged mice (red arrows) and placebo controls (not depicted) developed parasitemia (solid).
Point 3
Why for comparison was it not presented results of figure 1A (results obtained in vitro) parallel to the results pictured in figure 4 ( drug concentration in blood in vivo)? Why are both in vitro and in vivo releases completely different? In the data presented from the study of changes in the drug concentration in the blood (Fig. 4), depending on the time after the implantation of the depot, the statistical data should be processed and introduced to the chart (average standard deviation, confidence interval). Only after taking into account the data obtained from statistical calculations the course of the curve can be plotted.
Response 3:
In vitro release and in vivo exposure are quite different. In in vitro tests in a test tube, the drug released day by day has nowhere to go and is all accumulated in the tube. So it is described in the manuscript that any detected drug is the amount of drug released including all drugs released and accumulated from preceding times. All drugs do not have half-lives as those in animals or humans. The drug in the muscle released can be metabolized, transformed, or most likely excreted in the case of DQ and it has a half-life. So the drug detected each time only represents that released on the same day or recent days. So they are different scenarios and very hard to do the comparison.
For measuring the drug concentration in mice, the blood volume taken is very small. Each time point is one sample from one mouse. We take original data points to plot the graph. It is more straightforward and real. We did not do any modifications to the data. In fact, for delayed release formulation of drugs, all data measured in the blood tend to be at a low level and it is common practice to present data as they are.

Reviewer 2 Report
Most of my suggestions were addressed
Author Response
Thank you so much for your help. Your comments, critics, and suggestions improve the quality of the manuscript tremendously.